# An IT Operations Ontology for Knowledge-intensive Enterprise Use Cases

Rosario Uceda-Sosa[1], Nandana Mihindukulasooriya[1], Sahil Bansal[2], Seema Nagar[2], and Atul Kumar[2]

[1] IBM T.J. Watson Research Center, USA
[2] IBM Research, India

We present an Information Technology Operations ontology (ITOPS) that leverages Linked Open Data (LOD) resources as seeds for specialized industry ontologies. In our case, we needed a rich knowledge graph to support a range of applications related to IT Operations without investing in creating one from scratch. Some of the applications include multi-turn chatbots and a question-answering (QA) system that returns the relevant chats or text given a set of user keywords.

Since there was no publicly available ontology that could be used directly to support these applications [11] [1], and creating an ontology from scratch was too expensive ([6], [10], [7], [4]). our solution was to leverage general purpose LOD resources (such as Wikidata, Wikipedia and DBpedia) that include a wealth of well curated knowledge. However, due to their large size, complexity and the ambiguity generated in the search of technical terms, they could not be used by our applications. As an added concern, we needed to augment our ontology with proprietary user information.

Our solution was to implement a staged pipeline leveraging valuable LOD resources relevant to IT Operations. The overall resulting ontology, ITOPS, as well as each of the partial stages is publicly available for download and analysis[3].

An added advantage is that our pipeline, as described in the accompanying repository, is mostly automatic and has been used in a variety of other specialized industrial domains, like oil and gas, finance and healthcare.

Starting from a set of seed concepts (Figure 1), the first stage (S1) induces a Wikidata sub-graph including all their relevant relations (beyond IS-A and instanceOf) and instances. S2 adds extra concepts from richly annotated categories in Wikipedia articles for S1 terms. S3, adds proprietary terms through a model that analyzes the compatibility of the new terms with respect of those already in the ontology. Since we cannot share customer data, we've built our published version of S3 from IBM and Lenovo glossaries (details in GitHub).

**Applications**

ITOPS has already helped in several use cases such as generating facets to facilitate the search of customer support documents [8] or improving the performance of terminology extraction and ranking in knowledge induction scenarios ([5], [3], [9]). We're also exploring other uses, including powering an intelligent chatbot that leverages the entities and relations in the ontology to ask disam-

---

[3] https://github.com/IBM/ITOPS-ontology

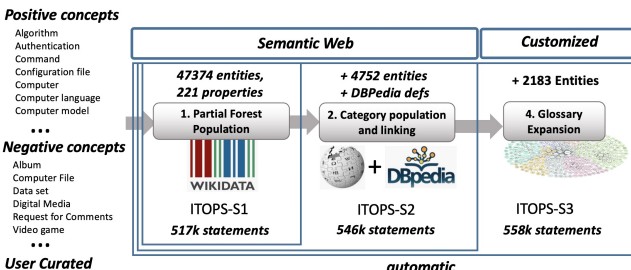

**Fig. 1.** Automated ontology construction pipeline from seed concepts

biguation questions in order to narrow down the best solution to the user's problem.

**Insights and Lessons Learned**

Being able to use Wikidata as a seed of specialized ontologies makes it a valuable resource for industrial applications. We tested the noise of the raw Wikidata in the IT domain by selecting the 500 most common IT Wikidata terms from the titles of the TechQA Dataset [2], which summarizes help requests about hardware, software or networking. Wikidata Term Search returned an IT term in places 1-3 only 15.6% and in places 1-5, 29.2% . For example, IT meanings of 'mask' appear in 13th place.

Another drawback of the general purpose Wikidata is the presence of (1) non productive, abstract concepts, like Variable Order Class (Q23958852), a result when querying the parents of, say, 'laptop', and (2) hierarchies that may be extraneous to our applications and bloat the graph, like instances of 'RFC' (Q212971), even though we may want the well known instances of its parent 'Technical Standard' (Q317623).

We have also used ITOPS to boost terminology ranking performance in a corpus of 4,000 technical troubleshooting (proprietary) documents. We used both labels and aliases in the ontology to re-rank the terms. For evaluation, we manually annotated a gold standard of the 480 most relevant terms. ITOPS helped to triple the average precision (from 4% to 12%) compared the state-of-the-art approach, CValue [12].

Another important need of technical ontologies in industry is the ability to absorb rapidly evolving vocabulary. Aligning with Wikidata, an active, curated knowledge graph, can simplify that task, especially if the process of inducing Wikidata subgraphs is automated. Keeping these ontologies in sync with Wikidata is not trivial, though. Strict provenance records must be kept about proprietary information, so it can be exported ahead of an update. This poses challenges as new Wikidata versions might alter the parent-child or other linkages of the proprietary concepts.

Despite the challenges, using Wikidata and Wikipedia as seed knowledge for specialized ontologies is casting them as an indispensable resource for industrial applications.

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
