# OpenReview forum: "An IT Operations Ontology for Knowledge-intensive Enterprise Use Cases"
_eswc-conferences.org/ESWC/2021/Conference/Industry_Track — Submitted to ESWC 2021 Industry_

### Official Review · ~Maria_Husmann1 · 2021-04-13
**Limited contribution compared to previous version, but potential for interesting insights into to use of public semantic data in an corporate context**

**Rating:** 7
**Confidence:** 3

**Review:**

The paper describes a (semi-) automated process of creating an ontology for IT operations based on publicly available linked open data, such as Wikidata. The paper is accompanied by a Github repository where the resulting ontology (but not the code for the pipleine) is stored.
An earlier version of the work has been published at a workshop ([link to pdf](https://kr2ml.github.io/2019/papers/KR2ML_2019_paper_44.pdf)). This previous work is currently not cited, but it should be as it provides more detail on the process.

The paper outlines applications and lessons learned, though remains superficial. Since this is the industry track with a very limited paper length, I understand that the authors could not elaborate on these topics on more in depth. For example, the authors write that "...[the pipeline] and has been used in a variety of other specialized industrial domains, like oil and gas, finance and healthcare." It is unclear, if the level of maturity of these applications is more at a proof-of-concept level or at actual product level. I would encourage the authors to set the focus on the applications and when they present their work at the conference. Concrete examples of the application of the ontology and the impact thereof would be helpful.

In summary, the paper has a limited contribution compared to its previous version. However, the project itself is interesting and a focus on applications and lessons learned during the conference would be a valuable contribution to the community.

---

### Official Review · ~Kimberly_Garcia1 · 2021-04-13
**An IT Operations Ontology for Knowledge-Intesive Enterprise Use cases**

**Rating:** 3
**Confidence:** 5

**Review:**

This work takes advantages of publicly available Linked data resources such as Wikidata and DBpedia to create an ontology for IT operations that could be reused in different applications.

Although the idea of deriving specialised ontologies from  existing resource is not new, this work is interesting given the possible applications. Unfortunately, the extended abstract lacks structure and clarity. Introducing and motivating the importance of the problem that is being tackled with the creation of an ontology is not only a common practice, but it is also essential to understand the objective of the work.

The second and third paragraphs of page 1 are somewhat contradictory; already in the second paragraph authors mention their "solution" but it is only to start more criticism over the current available linked data resources. Then, paragraphs 3 and 4 seem to be the core of the abstract but they are really vague. The reader is force to go to the repository, which present in more detail and in a  more fluent manner the proposed ontology.

Section "Insights and Lessons Learned" is a bit more structured but still does not clarify the  specific applications in which the ontology was used/tested.

Authors should clearly state the industrial applications that the proposed ontology has been used for and clarify their innovations and contributions.